# Patient participation in Delphi surveys to develop core outcome sets: systematic review

Heather Barrington  ,[1] Bridget Young,[2] Paula R Williamson[1]

## ABSTRACT

**Objectives** To describe the design and conduct of core outcome set (COS) studies that have included patients as participants, exploring how study characteristics might impact their response rates.

**Design** Systematic review of COS studies published between 2015 and 2019 that included more than one patient, carer or representative as participants (hereafter referred to as patients for brevity) in scoring outcomes in a Delphi.

**Results** There were variations in the design and conduct of COS studies that included patients in the Delphi process, including differing: scoring and feedback systems, approaches to recruiting patients, length of time between rounds, use of reminders, incentives, patient and public involvement, and piloting. Minimal reporting of participant characteristics and a lack of translation of Delphi surveys into local languages were found. Additionally, there were indications that studies that recruited patients through treatment centres had higher round two response rates than studies recruiting through patient organisations.

**Conclusions** Variability was striking in how COS Delphi surveys were designed and conducted to include patient participants and other stakeholders. Future research is needed to explore what motivates patients to take part in COS studies and what factors influence COS developer recruitment strategies. Improved reporting would increase knowledge of how methods affect patient participation in COS Delphi studies.

## Strengths and limitations of this study

► This is the first systematic review of patient participation in Delphi surveys for core outcome set development.
► This comprehensive review explored both study characteristics and recruitment and retention rates among patients.
► The findings are limited by reporting issues in the reviewed studies, especially on recruitment, and few studies reported how many individuals received the initial invitation to participate.
► Other reporting issues, including on patient and public involvement, limit the conclusions that can be drawn from this review.

## BACKGROUND

Patients and healthcare professionals need evidence about what treatments work best to inform their healthcare decisions. The results of clinical trials are, however, often difficult to compare due to a lack of standardisation in the outcomes measured for the same health condition and challenges with reporting bias.[1] In addition, including the perspectives of patients on what outcomes matter to them is crucial.[2] Core outcome sets (COS) are a potential solution to these problems, providing standardised sets of outcomes, developed and agreed on by key stakeholders, including patients.

COS are developed through iterative consensus building processes. Commonly, a systematic review and sometimes qualitative interviews with patients are used to explore patients' views on outcomes and generate a long list of potential outcomes. These outcomes are then taken forward into a consensus process, most gathering views through a Delphi survey and ratifying these results at a consensus meeting to agree on a COS.[3] Delphi participants are invited to score outcomes in several survey 'rounds', considering the feedback of other expert groups as part of the process. Delphi surveys lend themselves to e-surveys and as such can be widely distributed; however, like other questionnaires, these surveys are prone to low response rates.[4]

Patient participation in COS studies has increased over recent years, with Gargon *et al*[5] reporting 77% of published COS studies included patients or their representatives (eg, carers or patient advocates). While this paper focuses largely on patient participation in COS, it is important to distinguish between this and patient involvement in COS studies. When patients participate, they are contributing data on which outcomes to prioritise, for example, scoring outcomes in Delphi studies. When patients are involved in COS studies, they are helping to design and

[1]Department of Health Data Science, University of Liverpool, Liverpool, UK
[2]Department of Psychological Sciences, University of Liverpool, Liverpool, UK

**Correspondence to**
Heather Barrington;
Heather.Barrington@liverpool.ac.uk

oversee the COS study from a patient/public perspective. There are several challenges in including patient participants in COS, and indeed there are indications some COS developers 'problematise' patient participation,[6] highlighting, for example, the tendency for patients to rate many outcomes highly. Biggane et al[7] found that patients without prior experience of Delphi surveys expressed difficulty understanding both the purpose of the COS and particular aspects of the surveys. Young and Bagley[8] called for further research exploring how patient input is currently being sought in COS studies and to understand more about the challenges of including and engaging patients in COS development.

To the authors' knowledge, no review of patient participation in COS Delphi studies has previously been published. We have undertaken a systematic review of recent COS studies that have included patients in their COS Delphi, to describe how these studies have been designed and conducted and whether participation rates were linked with the study design variables: recruitment source, patient and public involvement (PPI) and reminders. By identifying challenges in recruiting and retaining patients in COS studies, this review aimed to inform strategies to optimise the participation of patients in future COS studies.

## METHODS
The protocol is available at: www.comet-initiative.org/Studies/Details/1824.

### Study selection
#### Inclusion criteria
Eligible COS studies were those identifying outcomes for use in research, published between 1 January 2015 and 31 December 2019 (to reflect current practice) and including more than one patient, carer or their representative as a participant (hereafter referred to as patients for brevity) in scoring outcomes in a Delphi as part of the process.

#### *Identification of relevant studies*
Studies were identified through the Core Outcome Measures in Effectiveness Trials (COMET) Initiative database. How studies are identified for inclusion in this database has previously been described.[3 5 9–13] Briefly, eligible studies for the database were those that employed methodology to gain consensus as to which outcome domains or outcomes should be measured in clinical trials or other forms of health research. Any studies that described the update of an existing COS are included in the database as linked papers to the original COS. Eligible studies are added to the database as they are identified, and an annual systematic review of these is published to ensure the database is kept current.

Studies meeting the criteria for our review were selected from the aforementioned database; linked studies were not included. Where authors referred the reader to the protocol in the methods section of their article, these protocols were also reviewed. Studies reporting updates to COS studies that were already in the COMET database were not included in the current review.

### Data extraction
A data extraction template was developed including the following domains:
► Study scope: health area, the population, intervention type and location (participating countries).
► Study development and design: methods to explore patients' views on outcomes; survey language and translation, and participant groups represented; number of rounds; number of outcomes in each round; reported PPI and piloting; scoring and feedback systems used; use of reminders and other incentives; and recruitment sources and methods.
► Study conduct and results: reporting of participant characteristics; response rates in each round by participant group; ratio professional experience (PE, ie, participants not providing a patient perspective, such as healthcare professionals and researchers): patients in round 1.

Some studies had included patients and other stakeholders earlier in their COS, for example, in generating a list of outcomes, and authors sometimes referred to these as 'rounds'. Only rounds relating to the scoring of outcomes were included in this review. Data extraction was undertaken by one person (HB) with checking of certain technical aspects, such as the methods of feedback, by a second person (PRW).

### Data analysis
In addition to describing how studies had been designed and conducted, we were keen to explore whether participation rates were linked with other study design variables. We anticipated, for example, that more personalised recruitment approaches or the use of incentives or reminders might impact response rates and that steps to enhance the design of surveys such as PPI and piloting might also impact patient participant responses. Additionally, we wished to explore whether the recruitment source used in a study influenced patient participation. The two most commonly used recruitment sources were patient organisations and treatment centres; therefore, these were chosen for comparison. As several studies used both these sources we also explored their combined influence on participation.

### Patient and public involvement
Patients and the public were not involved in the design, conduct or reporting of this review of previously published data.

## RESULTS
The Preferred Reporting Items for Systematic Reviews and Meta-Analyses diagram for the review is presented in

online supplemental figure 1. From a total of 146 COS studies published between 2015 and 2019, 73 COS studies were initially identified as eligible; however, two of these were subsequently excluded as only one patient had participated. Of the 71 included COS studies, 66 reported on a single COS. The remaining five studies reported on a total of 12 COS. For example, one article by Hall *et al*[14] reported on three COS for three different interventions in tinnitus. Patients could complete any or all of these Delphi surveys so recruitment and retention data for each of these COS studies could be different. After discussion, it was decided to treat each COS individually. Of the five articles that reported on more than one COS, two each reported on three COS, and three articles each reported on two COS. In total, therefore, 78 COS studies are included in this review. In 13 of the COS studies, patients participated in only one round of scoring in the Delphi.

### Study scope

Table 1 illustrates the scope of the included studies. The COS studies represented a broad range of health areas, with pregnancy and childbirth (14%, n=11) and cancer (12%, n=9) being the most common. While the COS were predominantly developed for adults (58%, n=45), 14% (n=11) were for children. Most COS were developed for any intervention (63%, n=49). The median number of countries participating in the COS studies was 16 (in 18 studies, the number of countries was either not reported or unclear), maximum 73 and 13% (n=10) were conducted in a single country. Where data were given for numbers of countries from which the patient participants were drawn, the maximum number of countries was 21.

### Study characteristics

The variation in study characteristics can be seen in table 2. In preparation for the Delphi study, the most common method used to explore patients' views on outcomes was by interview (n=20, 26%).

Thirty-six per cent of studies (n=28) described piloting the Delphi, while patient involvement in the study design or delivery provided in the main COS report was reported by 40% of studies (n=31), although the detail around the PPI and piloting was generally minimal.

Most COS studies were delivered electronically to patients (74%, n=39) and 59% (n=23) of these were delivered using the DelphiManager software developed by the COMET Initiative. Of the 51 studies that either reported on language used or where it was implicit in the description, 20% (n=10) of studies described offering some form of translation of the study materials (including three COS studies in one article). Just over half the studies reported using reminders (56%, n=44). Only 8% (n=6) of studies described using incentives, three monetary incentives and three non-monetary (three COSs from the same article).

A range of recruitment sources were used to recruit patients, and some studies used multiple sources. Patient organisations (62%, n=43) and treatment centres (45%, n=31) were the most common. The most common

| Table 1 | Scope of the core outcome set |
| --- | --- |
| **Core outcome set scope** | **n (%)** |
| Health area | |
| Anaesthesia and pain control | 1 (1) |
| Blood disorders | 1 (1) |
| Cancer | 9 (12) |
| Cancer/child health | 1 (1) |
| Child health | 1 (1) |
| Child health/ ear, nose and throat | 1 (1) |
| Child health/gastroenterology | 1 (1) |
| Ear, nose and throat | 4 (5)* |
| Endocrine and metabolic | 3 (4) |
| Eyes and vision | 1 (1) |
| Gastroenterology | 6 (8) |
| Healthcare of older people | 2 (3) |
| Heart and circulation | 3 (4) |
| Heart and circulation and skin | 3 (4)* |
| Kidney disease | 2 (3) |
| Lungs and airways | 2 (3) |
| Mental health | 1 (1) |
| Neonatal care | 1 (1) |
| Neurology | 4 (5)† |
| Neurology/eyes and vision | 1 (1) |
| Orthopaedics and trauma | 6 (8) |
| Other | 2 (3) |
| Overweight/obesity | 1 (1) |
| Pregnancy and childbirth | 11 (14)‡ |
| Rehabilitation | 1 (1) |
| Rehabilitation and rheumatology | 1 (1) |
| Rheumatology | 2 (3) |
| Skin | 4 (5) |
| Tobacco, drugs and alcohol dependence | 1 (1) |
| Adults/children | |
| Adults | 45 (58) |
| Both adults and children | 18 (23) |
| Children | 11 (14) |
| Not reported | 4 (5) |
| Gender | |
| Male only | 2 (3) |
| Female only | 8 (10) |
| Both | 68 (87) |
| Intervention | |
| Any | 49 (63) |
| Drug | 4 (5) |
| Psychological | 3 (4) |
| Surgery | 7 (9) |
| Other§ | 15 (19) |

Continued

| Table 1 Continued | |
|---|---|
| **Core outcome set scope** | **n (%)** |
| Countries (all participants) | |
| 1 only | 11 (19) |
| 2–10 | 14 (24) |
| 11–10 | 8 (14) |
| 20–30 | 13 (22) |
| >30 | 13 (22) |
| Not reported/unclear | 19 |

*Includes articles reporting three COS studies.
†Includes articles reporting two COS studies.
‡Includes two articles reporting two COS studies.
§Other: active surveillance anaesthetic techniques; behavioural; chemoradiotherapy; extracorporeal membrane oxygenation (ECMO); gene therapy; haemodialysis; healthcare transition; interdisciplinary multimodal pain therapy; medication review; physical activity intervention; prepregnancy care; procedure (induction of labour); rehabilitation; sound-based interventions; and visual screening/assessment.

method of recruitment was by email (74%, n=42). Online supplemental table 1a presents the data on professional recruitment sources and methods.

There was heterogeneity in reporting of patient participant characteristics. Only 10% (n=8) reported on the patient socioeconomic/educational status and only 9% (n=7) on their ethnicity. Similarly, less than a third of studies reported on either patient experience of the condition (eg, length of experience) or an aspect of their treatment experience. Table 5 presents the reporting data on professional characteristics. Additional study design characteristics are presented in online supplemental table 1b, and study characteristics relating to professionals are in online supplemental table 1a.

Table 3 presents the data on Delphi specific issues, including the duration of rounds, the scoring approaches in round 1 and feedback methods in round 2 (data for subsequent rounds are presented in online supplemental table 2a) where both patients and professionals scored outcomes. Most studies did not report the duration of their rounds; however, of those that did, the majority reported 2–4 weeks duration per round. The majority of COS studies reported using a 1–9 scoring system (70%, n=52).

Feedback methods were explored for studies reporting more than one round. Forty-eight studies reported on which stakeholder groups' feedback was presented to participants, for example, whether patient and professional feedback was presented separately for each group or combined. The most frequent approach was where results for different stakeholder groups were reported separately (n=21, 44%). A range of feedback types were described by the 43 studies reporting on this, with some studies reporting use of two or more types of feedback. The most common type of feedback was the distribution

| Table 2 Study characteristics of the Delphi studies | |
|---|---|
| **Study characteristics** | **n (%)** |
| **Methods to explore patients' views on important outcomes prior to the Delphi study*** | |
| Patient interviews | 20 (26) |
| Survey | 12 (12)† |
| Nominal group technique | 3 (4) |
| Focus groups | 4 (5) |
| Not reported/unclear | 47 |
| **Pilot Delphi undertaken** | |
| Pilot study reported | 28 (36)‡ |
| **Patient and public involvement (PPI)** | |
| PPI reported | 31 (40) |
| **Method of delivery (LE)** | |
| Electronic | 39 (74) |
| Post | 4 (8) |
| Face to face | 3 (6) |
| Mixture of approaches | 7 (13) |
| Not reported | 19 |
| Unclear | 6 |
| **Reminders** | |
| One reminder between rounds | 10 (31) |
| More than one reminder between rounds | 22 (69) |
| Reminders sent but number of reminders not reported | 12 |
| Not reported | 46§ |
| **Incentives (patient participants)** | |
| Yes (monetary incentive/voucher) | 3 (38) |
| Yes (non-monetary incentive)¶ | 3 (38) |
| Incentive not offered | 2 (25) |
| Not reported | 70 |
| **Language used with patients** | |
| Translation | 10 (20) |
| Conducted in English (specifically stated) | 19 (37) |
| Native language (implicit) | 22 (43) |
| Not reported | 27 |
| **Participant recruitment source and approach**** | |
| Recruitment source (patients) | |
| Patient organisation | 43 (62) |
| Clinic/treatment centre | 31 (45) |
| Social media | 19 (28) |
| PPI group (external to the COS study) | 14 (20) |
| Contacts of steering committee/management group | 7 (10) |
| Snowball sampling | 10 (15) |
| Research database | 6 (9) |

Continued

| Table 2 Continued | |
|---|---|
| **Study characteristics** | **n (%)** |
| Other†† | See footnote |
| Unclear‡‡ | 3 |
| Not reported | 6 |
| **Recruitment approach (patients)** | |
| Email invitation | 42 (74) |
| Postal invitation | 5 (9) |
| Telephone invitation | 4 (7) |
| Information provided in clinic | 7 (12) |
| Poster/newsletter | 7 (12) |
| e-source (website/social media) | 15 (30) |
| Recruitment approach unclear | 5 |
| Not reported | 16 |
| **Participant characteristics reported** | |
| **Patient participants** | |
| Age | 39 (50)§§ |
| Gender | 44 (56)¶¶ |
| Socio-economic/education | 8 (10)*** |
| Ethnicity | 8 (10)††† |
| Marital status | 7 (9) |
| Experience of condition | 24 (31) |
| Experience of treatment | 15 (19) |
| Other‡‡‡ | See footnote |

*Some studies used more than one approach to explore patients views on outcomes prior to the Delphi.
†Including six studies in which patients identified outcomes in what the authors referred to as 'round 1'.
‡Including three studies where pilots were without patients.
§Including 12 studies where reminders were sent but the number of reminders was not reported.
¶All non-monetary were certificates and reported in a single article.
**More than one recruitment source/approach may have been used.
††Other included through a professional organisation (n=2), a conference attended by patients (n=3, three COSs from the same article), previous participation in a research study (n=4) and participating researchers identified patients (n=1).
‡‡Additional articles partially unclear, recruitment source (n=3), recruitment approach (n=3).
§§Including five studies where age was reported collectively for both patients and professionals and one study where age reported for parent's child only.
¶¶Including 12 where COS study was specifically targeted at one gender and nine studies where gender was reported collectively for both patients and professionals.
***Including one study where education was reported collectively for both patients and professionals.
†††Including two studies where ethnicity was reported collectively for both patients and professionals.
‡‡‡Other: previous participation in research (n=2, both of which reported collectively for both patients and professionals), number of children (n=1) and home type (n=1).
COS, core outcome set; LE, lived experience.

of scores (65%, n=28); 10 studies (23% of those reporting) described providing either a mean or median only.

Table 4 shows the response rates per round. The recruitment sources of the 20 studies where patient response data for round 1 was reported were predominantly treatment centres (45%, n=9). The median round 1 response rate for patients was 59% compared with 52% for professionals. The median ratio of professionals to patients was 2.7 (n=61), although some studies reported more than twice as many patients as professionals (eg, Potter et al[15]).

Participation rates for rounds 2 and 3 were calculated (excluding studies where non-respondents were invited from previous rounds). The median round 2 response rate for patients was 84% (n=44), comparable with the professional respondents (median=85%, n=46). Response rates in round 3 were the same (91%) for both patients and professionals.

Table 5 explores potential associations between patient response rates, and PPI, Delphi piloting, reminders and methods of recruitment. There is limited reporting of data on these factors with no evidence of an effect of PPI, piloting and reminders on response rates but an indication that recruiting from treatment centres is better in terms of retention in round 2. Round 2 response rates for studies recruiting through treatment centres were higher (89%, n=6) than studies recruiting through patient organisations (77%, n=20) and a combined treatment centre/patient organisation approach (77%, n=11), although the numbers of studies were small, particularly for those recruiting through the treatment centre.

## DISCUSSION

This review has highlighted variations in the design and conduct of COS studies that included patients in the Delphi process, including differing: scoring and feedback systems, approaches to recruiting patients, lengths of time between rounds, and use of reminders, incentives, PPI and piloting. It has also identified potential challenges with the Delphi feedback approaches; minimal reporting of participant characteristics; the lack of translation of Delphi surveys into local languages; and indicated that recruitment may be more of a challenge than retention. There were indications that studies that recruited patients through treatment centres had higher round 2 response rates than studies recruiting through patient organisations.

### Previous qualitative research, PPI and piloting

Williamson et al[16] recommend using qualitative research or consulting with key stakeholders, including patients, to help identify important outcomes and ensure that the language used to describe outcomes is meaningful for patients. Less than a third of studies used either of these two methods prior to undertaking their Delphi survey. Additionally, Williamson et al[1] suggest that piloting of the Delphi survey can also help the COS development team to refine their outcome labels and explanations; however, only around a third of studies report undertaking piloting.

**Table 3** Delphi specific survey issues

### Duration of rounds

| Round duration | n (%) | | | |
|---|---|---|---|---|
| Time for each round | <2 weeks | 2–4 weeks | >4 weeks | Not reported/not clear/n/a |
| Round 1 | 1 (3) | 23 (70) | 9 (27) | 45 |
| Round 2 | 1 (3) | 25 (78) | 6 (19) | 46 |
| Round 3 | 0 | 16 (80) | 4 (20) | 58 |

| Scoring systems and feedback approaches | n (%) |
|---|---|
| **Scoring system (round 1)** | |
| 1–9/1–10* | 52 (70)† |
| 0–4/1–4/1–5 | 12 (16) |
| 9/10/12 most important outcomes | 4 (5) |
| Yes/no/don't know or agree/disagree/unsure | 7 (9) |
| Not reported | 2 |
| Unclear | 1 |
| **Source of stakeholder feedback round 2** | |
| All stakeholder groups combined | 10‡ (21) |
| Stakeholder groups reported separately | 21 (44) |
| Own stakeholder group only | 10§ (21) |
| Stakeholder groups reported separately and all stakeholder groups combined | 5 (10) |
| SWAT¶ – different groups saw different feedback | 1 (4) |
| N/a patients only took part in one round | 13 |
| Not reported | 13 |
| Unclear | 4 |
| **Feedback type reported**** | |
| Graphical feedback†† | 17 (40) |
| Numerical frequencies | 24 (56) |
| Summary statistics†† | 15 (35)‡‡ |
| Dispersion/distribution of scores | 28 (65) |
| Anonymised comments from prior round | 2 (5) |
| N/a patients only voted in one round | 13 |
| Not reported | 22 |

*Only two studies used 1–10.
†Children in one of these studies used 1–3 scale, and caregivers in another study scored differently to patients in one of these studies – patients used score cards.
‡Including one study that also provided the patient group scores and one study in which participants could request feedback by stakeholder group.
§Including one study which also provided combined scores for all.
¶Study Within a Trial.
**Studies could report more than one type of feedback.
††Excludes anywhere it was unclear whether the feedback type was reported.
‡‡10 studies reported only summary statistics.

COS developers may be missing opportunities to improve the accessibility of their Delphi surveys. Better reporting of piloting would improve understanding of its impact.

Young and Bagley[8] described the potential benefits that PPI could bring to the COS development process. PPI has the potential, for example, to help with recruitment and retention by improving the accessibility of the study. Less than half of the publications in this review reported undertaking PPI; those that did report PPI provided scant details. It is acknowledged that word restrictions and the journal's focus may limit the amount of space that can be dedicated to discussions about PPI and also that some authors may

**Table 4** Response rates

| Round | Participation | Median, min, max |
|---|---|---|
| 1 | Patients invited and completed (n=20) | 59%, 11%, 95% |
| | Professionals invited and completed (n=20) | 52%, 19%, 93% |
| | Ratio of professionals to patients (n=62) | 2.7, 4.1, 0.4, 23 |
| 2 | Patients invited and completed (n=44) | 84%, 32%, 100% |
| | Professionals invited and completed (n=46) | 85%, 43%, 100% |
| 3 | Patients invited and completed (n=20) | 91%, 50%, 100% |
| | Professionals invited and completed (n=24) | 91%, 78%, 100% |

*In round two and / or round three some studies described non-responders to a previous round being invited into the round (this could be both patient and professional previous responders or just one type of previous responder). These studies were excluded from analysis of round two and / or round three response rate data for the relevant category of respondent. Round one participation rates were available for studies where the denominator was known (ie, the number of people invited).

**Table 5** Association between patient response rate and PPI, piloting and recruitment source

| Factor | Round* | Factor category | Patients – median response rate, min, max |
|---|---|---|---|
| PPI | 1 | PPI (n=6) | 62%, 36%, 77% |
| | | PPI not reported (n=14) | 59%, 11%, 95% |
| | 2 | PPI (n=22) | 78%, 32%, 94% |
| | | PPI not reported (n=22) | 86%, 50%, 100% |
| | 3 | PPI (n=9) | 92%, 71%, 100% |
| | | PPI not reported (n=11) | 90%, 50%, 100% |
| Piloting | 1 | Piloting (n=10) | 61%, 36%, 95% |
| | | No piloting reported (n=10) | 58%, 11%, 91% |
| | 2 | Piloting (n=21) | 84%, 41%, 100% |
| | | No piloting reported (n=23) | 83%, 32%, 100% |
| | 3 | Piloting (n=9) | 92%, 71%, 100% |
| | | No piloting reported (n=11) | 89%, 50%, 100% |
| Recruitment source | 2 | Treatment Centre (n=6) | 89%, 83%, 90% |
| | | Patient organisation (n=20) | 77%, 32%, 100% |
| | | Treatment centre and patient organisation (n=11) | 77%, 50%, 93% |
| | | Neither treatment centre nor patient organisation (n=5) | 94%, 90%, 100% |
| | | Nothing reported on recruitment source (n=2) | 92%, 84%, 100% |
| Reminders | 2 | Reminders (n=30) | 82, 32,96 |
| | | No reminders reported (n=14) | 86, 57, 100 |

*Nineteen studies with round 1 data on participation rate, 44 studies with round 2 completion rate and 20 with round 3 completion rate data.
PPI, patient and public involvement.

have chosen to publish separately about PPI in their COS studies, for example, Smith et al.[17] This review did not include linked papers to the COS studies, and this, therefore, limits the conclusions that can be drawn; however, the experience of the COMET Initiative suggests that such detailed publications about PPI in COS and its impact are rare. The few studies that did provide more detailed reports will help future COS developers plan for PPI (eg, Smith et al[17] and Crudgington et al[18]). Improving the reporting of PPI, for example, by following the GRIPP2 checklist,[19] would enable the impact of PPI on recruitment and retention to be more accurately investigated.

We explored the potential impact of PPI on patient participation rates but did not find an association. Minimal reporting of PPI however means that it was also unclear what the quality of PPI was like, for example, one study might have held multiple supported meetings with a number of patients to explore how to define the outcomes for a study, where another study might only have emailed a list of outcomes for feedback from one research partner, with little guidance on how to review the outcomes for a patient audience. Without such detail, it is difficult to come to conclusions about the real impact of PPI. Ethnographic work with patient research partners in COS studies will inform our understanding of current PPI practice in this area[20]

### Scoring system and feedback

Our review indicates that the 1–9 scoring system is the most commonly used system in COS studies that include patients; however, this scoring system is used in the DelphiManager software, and the large number of electronically delivered studies that reported using this software may, therefore,

have influenced this finding. Biggane et al[7] interviewed patients retrospectively about their experience of participating in a Delphi survey, noting that while there are statistical considerations influencing the choice of scoring scales, patients can have differing views on the scales used. While some patients in their study preferred the 1–9 scoring scale, others struggled to use it, indicating the need for additional support and guidance. Given the high usage of the nine-point scoring method, further research is warranted to explore how patients and other participants experience, interpret and use this scoring system.

Providing feedback to participants on the scores of other participants in previous rounds is used to drive consensus between stakeholders in Delphi surveys, with stakeholders encouraged to consider the views of others before rescoring an outcome. A study that compared providing feedback to participants only on the scores of their own peer group, versus providing feedback to participants on the scores from each of the stakeholder groups, found that seeing other groups' perspectives increased consensus.[21] Participants in a study by Fish et al[22] reported 'trying to understand the importance of an outcome from the perspective of another participant', as one of the most common reasons for revising their scores between rounds, and this was especially the case for healthcare professionals. While several studies in our review did not report on their feedback approach, nearly half of those that did report this did not describe providing feedback to participants by group, instead just presenting feedback from a participant's own stakeholder group or for all participants combined. In the absence of presenting each participant with feedback from each group, consensus may not be so easily achieved across stakeholder groups.[1] Of note were two SWAT studies exploring feedback methods, indicating interest in finding the best feedback approach.[23 24] One of these has been completed, finding that peer feedback reduced variability in scoring compared with combined feedback from multiple groups.[23] It should again be noted that the use of DelphiManager software by a large proportion of studies conducted electronically may have impacted the data on feedback.

In addition to what feedback participants received about the scores of other participants, how feedback was presented also varied in the studies although most presented feedback as a distribution of scores and numerical frequencies. Of studies that reported on how feedback was presented, a fifth described only providing a summary statistic (a median or mean score). This is potentially problematic as there are indications that participants do not understand the term median and that they have issues with fully understanding averages.[25] Fish[25] also found the patients in her study understood and liked seeing the percentage of participants rating each outcome as each of 1–9, and yet our review has found that around two-thirds of studies did not provide such feedback. Further research is needed to explore the best ways to present feedback so that it is more easily understood.

### Patient participation and inclusivity

The COS-STAD (Standards for Core Outcome Set Development) specifies that people with lived experience of the condition/intervention should be key stakeholders in the COS development process.[26] Our review explored the ratio of patient participants compared with professionals, finding that patients tended to be in the minority, although there are also examples of COS studies with higher rates of participation among patients (eg, Potter et al[21]). Inclusivity in COS development is crucial to ensure that the outcomes selected in a COS are relevant and important for the diverse range of patients potentially affected by the COS. There have been calls for more inclusive research generally,

further emphasised by the recent COVID-19 pandemic.[27] In the studies in our review, there was minimal reporting of patient ethnicity and socioeconomic status, and the reasons for this warrant further exploration. Additionally, there was minimal reported use of translation meaning that COS completion is restricted to those with the relevant language skills, again limiting its inclusivity.

Given the need to ensure adequate stakeholder diversity and inclusion and the potential impact of attrition (overestimation of consensus if participants with minority perspectives drop out), it is important to explore response rates in all rounds of the COS studies. There are indications that recruiting stakeholder participants into COS studies can be challenging; however, once recruited, retention was quite high for most studies. This echoes findings from Delphi studies in other areas.[4] Retrospective interviews with patient participants in COS Delphi studies have highlighted key areas of concern for them and provided some initial insights on their motivation to participate.[7] However, further research is needed that explores patients' motivation to take part soon after the recruitment decision to inform the development of future recruitment resources.

### Associations with patient participation rates

We aimed to explore how study characteristics such as PPI, piloting, reminders, recruitment methods and sources influenced the participation of patients. The reporting of recruitment in the reviewed studies was complex and sometimes unclear. Our comparison of recruitment sources and response rates was limited due to problems with reporting. However, studies using treatment centres as a source for recruitment appeared to have higher round 2 response rates. This echoes previous findings[25] indicating lower attrition among patient participants recruited via treatment centres compared with those recruited through patient organisations and social media. This warrants further research.

### Study limitations and future research

This study is limited by omissions in reporting about the design and delivery of studies. Recent guidance about COS development and reporting[28] and guidance on PPI reporting[19] may improve the description of COS studies in the future. We are planning to interview COS developers to explore their perspectives on the design of COS Delphi studies, including the use of patient facing resources to recruit and retain patients in a Delphi survey and materials to support their participation. We will work closely with a PPI panel to review these materials, alongside the findings of this current review and the future findings from interviews with COS developers, to enhance the accessibility, ease of use and appeal of the materials.

### CONCLUSION

This study has explored the participation of patients in COS studies. Variability was striking in how COS Delphi surveys were designed and conducted to include patient

participants and other stakeholders. Future research would be useful to explore what motivates patients to take part in COS studies and what factors influence recruitment strategies used by COS developers. Reporting needs to be improved to increase knowledge of how methods affect patient participation, in particular reporting response rates and denominators for all rounds by stakeholder group, more detailed descriptions of PPI, piloting, recruitment methods and sources.

**Contributors** Conceptualisation: all authors; funding acquisition: PRW; investigation: all authors; methodology: all authors; writing – original draft: HB; writing – review and editing: all authors.

**Funding** HB is supported by the National Institutes for Health Research (NIHR) through award number NF-SI_0513–10025.

**Disclaimer** The views expressed in this article are those of the author(s) and not necessarily those of the NIHR, or the Department of Health and Social Care. PW is also supported by the Medical Research Council Trials Methodology Research Partnership (grant reference MR/S014357/1).

**Competing interests** PRW and HB are members of the COMET Management Group; BY and HB are members of the COMET PoPPIE Working Group.

**Patient consent for publication** Not required.

**Provenance and peer review** Not commissioned; externally peer reviewed.

**Data availability statement** Data are available on reasonable request. Date are available upon reasonable request from the first author.

**ORCID iD**
Heather Barrington http://orcid.org/0000-0001-9103-2670

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
