## [Reviewer comments · BMJ Open]

ARTICLE DETAILS

TITLE (PROVISIONAL)	Patient participation in Delphi surveys to develop core outcome sets: systematic review
AUTHORS	Barrington, Heather; Young, Bridget; Williamson, Paula

VERSION 1 – REVIEW

REVIEWER	Ann Hall, Deborah University of Nottingham
REVIEW RETURNED	25-Apr-2021

GENERAL COMMENTS	This is an interesting and valuable review describing how COS studies have been designed and conducted and how study characteristics might impact patient participant response rates. Page 2 line 54: “...how study characteristics might impact patient participant response rates” could be rephrased to be more descriptive: “...whether participation rates were linked with other study design variables such as recruitment source, personalised recruitment approaches, incentives and reminders.” Page 3 lines 15-28 : The authors state that “patient involvement in the study design or delivery was reported by 40% of studies (n=31), although the detail around the PPI and piloting was generally minimal”. Did the authors consider whether the details of the PPI could be reported in more detail in other linked publications that deal with PPI rather than the outcome consensus per se? The methods described in “Identification of relevant studies” does not clearly answer this question. For example, I am not sure how to interpret the sentence: “Any studies that described the update of an existing COS are included in the database as linked papers to the original COS”. If the authors did gather all those additional articles reported on the PPI aspects of the COS development, then it would v=be helpful to summarise that information so its readily available to the reader. Due to word limits and journal focus, detailed descriptions of the PPI methods and impacts are not always appropriate to the article reporting the COS development per se. And therefore it would seem appropriate to widen the study identification to include these associated articles. This was certainly true in my experience with COMiT, and my team has published separate peer reviewed journal papers detailing the PPI methods and evaluating their impacts.
--

	If the authors did not include these linked publications in their systematic review evidence synthesis, then a caveat is required to explain this since the implicit criticism may be a little too harsh. Results If the authors did include these linked publications in their systematic review it would be helpful to give more details in the results section. For example, of the 40% of COS studies reporting PPI, how many only reported PPI methods in the COS final report, versus in the protocol versus independent publications that focused on questions around the PPI itself. Page 4 line 24: “The majority of COS studies reported using a 1-9 scoring system (70%, n=52).” This is one of the findings that was most consistent across included studies. To what extent is this due to the studies using the same software platform (e.g. DelphiManager) to gather electronic data? It would be informative to examine and report since this consistency in method could then be attributed to the positive influential role of the COMET project in the field. Other common methods such as reporting feedback as a distribution of scores (65%) might also be explained in the same way. Discussion While paragraph 1 summarises the usual main findings and conclusions, the Discussion section does not have any subheadings and so I found it difficult to appreciate its structure and organisation. Paragraph 2 (Page 6 line 15-24) seems to be somewhat incidental. For example, while Williamson et al. [16] recommended using qualitative research or consulting with patients to help identify important outcomes and ensure that the language used to describe outcomes is meaningful for patients, why then did PPI not influence patient participation rates? Paragraph 2 should instead focus on considering why PPI did not seem to generally play a beneficial role in study outcomes. And the authors might consider if this is the finding they really want to emphasise first (since it’s effectively a null finding). I suggest to move up the discussion on PPI reporting (Page 7 line 38-45). Paragraph 3 (Page 6 line 26-34) limits itself to discussing the 1-9 rating scale. However, see my comments above about reframing the issue by grouping together discussion of those methods elements that may be driven by the choice of software platform. The topic of feedback methods (Page 6/7 line 26-line 09) is also relevant here.
--	---

REVIEWER	Alam, Murad Northwestern University Feinberg School of Medicine, Dermatology
REVIEW RETURNED	02-May-2021

GENERAL COMMENTS	Dear Authors, Thank you for addressing this important topic. Better characterizing patient involvement in COS studies is critical to understanding how this can be improved in order to better represent the patient perspective in such work.
--

	You may wish to consider the following minor modifications:  - Tables 1-3: not all data points are reported as n (%). It may be helpful to ensure this is consistent in all cells. - Pg. 1, line 34: the comma is not necessary between "including, differing" - Pg. 2, line 28: perhaps a word is missing: "consensus meeting to agree [upon] a COS" - Pg. 2, line 35: it may be helpful to define, or list examples, or a patient representative. - Pg. 4, lines 24-25: Consider moving the following from Results to Methods: "After discussion it was decided to treat each COS individually" - Pg. 4, line 44: delete "o" from "...in the description, o 20%" Thank you again for your submission and for the opportunity to review your work. I look forward to viewing the final publication.
--	--

REVIEWER	Blackwood, Bronagh Queen's University Belfast, Wellcome-Wolfson Institute for Experimental Medicine
REVIEW RETURNED	10-May-2021

GENERAL COMMENTS	Thank you for the opportunity to review this very well written manuscript systematic review addressing patient participation in Delphi surveys to develop core outcome sets. It is a novel review addressing an important aspect of core outcome set study design: inclusion of patients. The aims and objectives of the review are clear and results are clearly outlined. I believe the discussion and conclusions that the authors draw from this study are justified by the results provided. Review limitations and future directions for research in this area are also provided. The reference list is up to date and contains key references in this field. The supplementary material is clearly presented and easy to follow, with correct corresponding tables referenced throughout the manuscript. I recommend some very minor edits outlined below.  1. Please provide a rationale in the methods section for including COS studies from January 2015 -2019 and a reason why COS studies pre-dating 2015 were not included in their systematic review search strategy. 2. Some minor editing changes to the text are suggested below: Page 5, line 16: suggest replacing 'The PRISMA diagram for the review in presented etc' with 'The PRISMA diagram for the review is presented etc.' Page 5, line 55 remove typo 'o' Page 7, line 26: suggest replacing 'Our review indicates that the 1-9 scoring system is most commonly used system in COS studies etc' with 'Our review indicates that the 1-9 scoring system is the most commonly used system in COS studies etc'. Page 7, line 46: suggest rewording 'Whilst several studies in our review did not report on their feedback approach, nearly half of those that did report described providing reported on this did not provide feedback to participants by group etc' to 'Whilst several studies in our review did not report on their feedback approach, nearly half of those that did report this, did not provide feedback to participants by group etc'.
--

VERSION 1 – AUTHOR RESPONSE

Reviewer: 1

Dr. Deborah Ann Hall, University of Nottingham Comments to the Author:

This is an interesting and valuable review describing how COS studies have been designed and conducted and how study characteristics might impact patient participant response rates.

Page 2 line 54:

“...how study characteristics might impact patient participant response rates” could be rephrased to be more descriptive: “...whether participation rates were linked with other study design variables such as recruitment source, personalised recruitment approaches, incentives and reminders.” We were unable to explore associations for variables such as incentives due to the small numbers of studies using this approach so this sentence has been changed to: “...whether participation rates were linked with the study design variables: recruitment source, PPI and reminders.”

Page 3 lines 15-28 :

The authors state that “patient involvement in the study design or delivery was reported by 40% of studies (n=31), although the detail around the PPI and piloting was generally minimal”. Did the authors consider whether the details of the PPI could be reported in more detail in other linked publications that deal with PPI rather than the outcome consensus per se? The methods described in “Identification of relevant studies” does not clearly answer this question. For example, I am not sure how to interpret the sentence: “Any studies that described the update of an existing COS are included in the database as linked papers to the original COS”. If the authors did gather all those additional articles reported on the PPI aspects of the COS development, then it would v=be helpful to summarise that information so its readily available to the reader.

Due to word limits and journal focus, detailed descriptions of the PPI methods and impacts are not always appropriate to the article reporting the COS development per se. And therefore, it would seem appropriate to widen the study identification to include these associated articles. This was certainly true in my experience with COMiT, and my team has published separate peer reviewed journal papers detailing the PPI methods and evaluating their impacts.

If the authors did not include these linked publications in their systematic review evidence synthesis, then a caveat is required to explain this since the implicit criticism may be a little too harsh.

This is a really helpful observation, we didn't include linked studies, we have therefore now made this clear in the method section, highlighted this in the results and reflected on the limitation of this in the discussion section of the manuscript. See below:

Method - “Studies meeting the criteria for our review were selected from the aforementioned database, linked studies were not included. Where authors referred the reader to the protocol in the methods section of their article, these protocols were also reviewed. Studies reporting updates to COS studies that were already in the COMET database were not included in the current review”.

Results – “Thirty six percent of studies (n = 28) described piloting the Delphi, whilst patient involvement in the study design or delivery provided in the main COS report was reported by 40% of studies (n=31), although the detail around the PPI and piloting was generally minimal”.

Discussion - Young & Bagley[8] described the potential benefits that PPI could bring to the COS development process. PPI has the potential, for example, to help with recruitment and retention by

improving the accessibility of the study. Less than half of the publications in this review reported undertaking PPI; those that did report PPI provided scant details. It is acknowledged that word restrictions and the journal's focus may limit the amount of space that can be dedicated to discussions about PPI and also that some authors may have chosen to publish separately about PPI in their COS studies, for example, Smith[17]. This review did not include linked papers to the COS studies and this, therefore, limits the conclusions that can be drawn, however, the experience of the COMET Initiative suggests that such detailed publications about PPI in COS and its impact are rare. The few studies that did provide more detailed reports will help future COS developers plan for PPI (e.g. Smith[17] & Crudgington[18]). Improving the reporting of PPI, for example, by following the GRIPP2 checklist [19], would enable the impact of PPI recruitment and retention to be more accurately investigated.

Results

If the authors did include these linked publications in their systematic review it would be helpful to give more details in the results section. For example, of the 40% of COS studies reporting PPI, how many only reported PPI methods in the COS final report, versus in the protocol versus independent publications that focused on questions around the PPI itself. As above, we did not extract this level of detail, however, hopefully the revisions made to the discussion provide greater depth to our discussion about PPI in COS.

Page 4 line 24: "The majority of COS studies reported using a 1-9 scoring system (70%, n=52)." This is one of the findings that was most consistent across included studies. To what extent is this due to the studies using the same software platform (e.g. DelphiManager) to gather electronic data? It would be informative to examine and report since this consistency in method could then be attributed to the positive influential role of the COMET project in the field. Other common methods such as reporting feedback as a distribution of scores (65%) might also be explained in the same way.

Thank you for drawing our attention to this important observation. We have amended the Results and Discussion sections as follows:

Results – "Most COS studies were delivered electronically to patients (74%, n =39) and 59% (n=23) of these were delivered using the DelphiManager software developed by the COMET Initiative".

Discussion – "Our review indicates that the 1-9 scoring system is the most commonly used system in COS studies that include patients, however this scoring system is used in the DelphiManager software, and the large number of electronically delivered studies that reported using this software may, therefore, have influenced this finding".

"Of note were two SWAT studies exploring feedback methods, indicating interest in finding the best feedback approach[20,21]. One of these has been completed, finding that peer feedback reduced variability in scoring compared with combined feedback from multiple groups[20]. "It should again be noted that the use of DelphiManager software by a large proportion of studies conducted electronically may have impacted the data on feedback".

Discussion

While paragraph 1 summarises the usual main findings and conclusions, the Discussion section does not have any subheadings and so I found it difficult to appreciate its structure and organisation. Thank you for highlighting this. Subtitles now included in the discussion

Paragraph 2 (Page 6 line 15-24) seems to be somewhat incidental. For example, while Williamson et al. [16] recommended using qualitative research or consulting with patients to help identify important outcomes and ensure that the language used to describe outcomes is meaningful for patients, why

then did PPI not influence patient participation rates? Paragraph 2 should instead focus on considering why PPI did not seem to generally play a beneficial role in study outcomes. And the authors might consider if this is the finding they really want to emphasise first (since it's effectively a null finding). I suggest to move up the discussion on PPI reporting (Page 7 line 38-45).

Thank you for highlighting this, we can see the real benefit of doing this. The section on PPI reporting has been moved up including the restrictions from not reviewing linked articles. Additionally, we have added the following:

"We explored the potential impact of PPI on patient participation rates, but did not find an association. However, the minimal reporting means that it was unclear what the quality of PPI was like, for example, one study might have held multiple supported meetings with a number of patients to explore how to define the outcomes for the study, where another study might only have emailed a list of outcomes for feedback from one research partner, with little guidance on how to review the outcomes for a patient audience. Without such detail it is difficult to come to conclusions about the real impact of PPI on a COS study. Ethnographic work with patient research partners in COS studies will inform our understanding of current PPI practice in this area [20]".

Paragraph 3 (Page 6 line 26-34) limits itself to discussing the 1-9 rating scale. However, see my comments above about reframing the issue by grouping together discussion of those methods elements that may be driven by the choice of software platform. The topic of feedback methods (Page 6/7 line 26-line 09) is also relevant here. Addressed as highlighted earlier in relation to the comment about the software used.

Reviewer: 2

Dr. Murad Alam, Northwestern University Feinberg School of Medicine Comments to the Author:
Dear Authors,

Thank you for addressing this important topic. Better characterizing patient involvement in COS studies is critical to understanding how this can be improved in order to better represent the patient perspective in such work.

You may wish to consider the following minor modifications:

- Tables 1-3: not all data points are reported as n (%). It may be helpful to ensure this is consistent in all cells. We do not report the % for unclear/not reported responses since they are removed from the denominator for calculation of others %s
- Pg. 1, line 34: the comma is not necessary between "including, differing" Comma removed
- Pg. 2, line 28: perhaps a word is missing: "consensus meeting to agree [upon] a COS" The word upon has been added
- Pg. 2, line 35: it may be helpful to define, or list examples, or a patient representative. Example added "(for example, carers or patient advocates)"
- Pg. 4, lines 24-25: Consider moving the following from Results to Methods: "After discussion it was decided to treat each COS individually" We prefer to leave this decision in the Results section since it was made after identifying the studies.
- Pg. 4, line 44: delete "o" from "...in the description, o 20%" o deleted

Thank you again for your submission and for the opportunity to review your work. I look forward to viewing the final publication.

Reviewer: 3

Prof. Bronagh Blackwood, Queen's University Belfast Comments to the Author:

Thank you for the opportunity to review this very well written manuscript systematic review addressing patient participation in Delphi surveys to develop core outcome sets. It is a novel review addressing an important aspect of core outcome set study design: inclusion of patients. The aims and objectives of the review are clear and results are clearly outlined. I believe the discussion and conclusions that the authors draw from this study are justified by the results provided. Review limitations and future directions for research in this area are also provided. The reference list is up to date and contains key references in this field. The supplementary material is clearly presented and easy to follow, with correct corresponding tables referenced throughout the manuscript.

I recommend some very minor edits outlined below.

1. Please provide a rationale in the methods section for including COS studies from January 2015 - 2019 and a reason why COS studies pre-dating 2015 were not included in their systematic review search strategy. The inclusion criteria in the Methods has been changed to: "Inclusion criteria Eligible COS studies were those identifying outcomes for use in research, published between 1st January 2015 and 31st December 2019 (to reflect current practice), and including more than one patient, carer or their representative as a participant (hereafter referred to as patients for brevity) in scoring outcomes in a Delphi as part of the process".

2. Some minor editing changes to the text are suggested below:

Page 5, line 16: suggest replacing 'The PRISMA diagram for the review in presented etc' with 'The PRISMA diagram for the review is presented etc.' – Word in changed to is.

Page 5, line 55 remove typo 'o' o deleted

Page 7, line 26: suggest replacing 'Our review indicates that the 1-9 scoring system is most commonly used system in COS studies etc' with 'Our review indicates that the 1-9 scoring system is the most commonly used system in COS studies etc'. Word 'the' added

Page 7, line 46: suggest rewording 'Whilst several studies in our review did not report on their feedback approach, nearly half of those that did report described providing reported on this did not provide feedback to participants by group etc' to 'Whilst several studies in our review did not report on their feedback approach, nearly half of those that did report this, did not provide feedback to participants by group etc'. Sentence changed to "Whilst several studies in our review did not report on their feedback approach, nearly half of those that did report this, did not describe providing feedback to participants by group, instead just presenting feedback froa participant's own stakeholder group or for all participants combined".

Reviewer: 1

Competing interests of Reviewer: I have a professional connection to the authors through my membership of the MRC-NIHR Trials Methodology Research Partnership (TMRP).

I am the first author of one of the COS studies reported in the article.

Reviewer: 2

Competing interests of Reviewer: None.

Reviewer: 3

Competing interests of Reviewer: None